# Histopathological Markers for Target Therapies in Primary Cutaneous Lymphomas

**DOI:** 10.3390/cells12222656

**Published:** 2023-11-20

**Authors:** Benedetta Sonego, Adalberto Ibatici, Giulia Rivoli, Emanuele Angelucci, Simona Sola, Cesare Massone

**Affiliations:** 1Dermatology Clinic, University of Trieste, 35125 Trieste, Italy; benedetta.sonego@gmail.com; 2UO Ematologia e Terapie Cellulari, IRCCS Ospedale Policlinico San Martino, 16132 Genoa, Italy; adalberto.ibatici@hsanmartino.it (A.I.); giulia.rivoli@hsanmartino.it (G.R.); emanuele.angelucci@hsanmartino.it (E.A.); 3Surgical Pathology, Galliera Hospital, 16128 Genoa, Italy; simona.sola@galliera.it; 4Dermatology Unit & Scientific Directorate, Galliera Hospital, 16128 Genoa, Italy

**Keywords:** mycosis fungoides, primary cutaneous anaplastic large T-cell lymphoma, lymphomatoid papulosis, brentuximab vedotin, mogamulizumab, tagraxofusp, nivolumab, pembrolizumab, atezolizumab, lacutamab

## Abstract

In recent years, targeted (biological) therapies have become available also for primary cutaneous T-cell lymphomas (PCTCLs) including anti-CD30 (brentuximab vedotin) in mycosis fungoides, primary cutaneous anaplastic large T-cell lymphoma, lymphomatoid papulosis; anti-CCR4 (mogamulizumab) in Sezary syndrome; anti-CD123 (tagraxofusp) in blastic plasmocytoid cell neoplasm. Moreover, anti-PD1 (nivolumab), anti-PDL1 (pembrolizumab, atezolizumab), anti-CD52 (alemtuzumab), anti-KIR3DL2-CD158k (lacutamab), and anti-CD70 (cusatuzumab) have been tested or are under investigations in phase II trials. The expression of these epitopes on neoplastic cells in skin biopsies or blood samples plays a central role in the management of PCTCL patients. This narrative review aims to provide readers with an update on the latest advances in the newest therapeutic options for PCTCLs.

## 1. Introduction

Primary cutaneous lymphomas (PCLs) are a heterogeneous group of extranodal non-Hodgkin lymphomas that manifest exclusively in the skin. PCL can arise from neoplastic T and B-cells, with cutaneous T-cell lymphomas (CTCLs) being more frequent, accounting for approximately 75–80% of all PCLs. Among CTCLs, mycosis fungoides (MF) represent about 60% of CTCL cases and approximately 40% of all PCLs. Primary cutaneous CD30+ lymphoproliferative disorders (pcLPDs) make up around 30% of CTCL cases [1,2,3,4].

MF and pcLPDs (including pc anaplastic large cell lymphoma, ALCL, and lymphomatoid papulosis, LyP) typically follow an indolent clinical course, with a 5-year disease-specific survival (DSS) of 88% and 99%, respectively [1,2,3]. In contrast, Sezary syndrome (SS) is a rare leukemic type of CTCL, accounting for about 2% of cases, and has a 5-year DSS of 36% [1,2,3]. Even more aggressive is blastic plasmacytoid dendritic cell neoplasm (BPDCN), a rare hematologic neoplasm (less than 1% of primary cutaneous lymphomas and acute leukemia) that originates from precursors of the type 2 or plasmacytoid.

Dendritic cells (pDCs), often presenting with skin, bone marrow, and central nervous system (CNS) involvement [5,6,7,8,9,10,11].

Cutaneous B-cell lymphomas (CBCL) encompass indolent variants (such as primary cutaneous marginal zone lymphoma, pcMZL, and primary cutaneous follicle center lymphoma, pcFCL) as well as an intermediate type (primary cutaneous large B-cell lymphoma, leg type, pcDLBCL, LT). The 5-year disease-specific survival (DSS) rates for these subtypes are 95% and 56%, respectively [2,3,12].

Until a few years ago, treatment options for patients with progression to tumor stage MF or SS, diffuse ALCL, and BPDCN were limited, primarily involving radiotherapy or various chemotherapeutic regimens [13]. However, new molecular therapies have recently become available. Notably, these therapies target molecular antigens expressed by neoplastic lymphocytes, serving as both therapeutic targets and diagnostic histopathological markers [11,14,15,16,17].

This narrative review will focus on the latest therapies and advancements for MF, SS, ALCL, LyP, and BPDCN (Table 1). CTCLs other than MF, SS, and pcLPDs, as well as treatment of CBCL and other rare variants of CBCL, are not discussed in this review.

## 2. Principles of Histopathology, Immunohistochemistry, and Biomarkers of CTCL and BPDCN

### 2.1. Histopathology of the Patch-Plaque Stage of Mycosis Fungoides

MF is the prototypic epidermotropic lymphoma. Neoplastic lymphocytes are detected in the epidermis in 96% of biopsies during the patch stage of MF. These epidermotropic lymphocytes are often found as single units through the epidermis or along the basal layer. In approximately only 20% of cases, they are collected in the typical Pautrier’s (Darier’s) microabscess. Atypical lymphocytes are diagnostically significant but are rarely observed in early MF, occurring in only about 10% of biopsies. Interestingly, a band-like infiltrate is present in 30% of cases, while a patchy lichenoid infiltrate is seen in 66% of cases in the superficial dermis. In the plaque stage of MF, similar features to those in early MF are observed, but there is more pronounced epidermal hyperplasia and a more prominent dermal infiltrate. Moreover, Pautrier’s (Darier’s) microabscesses and atypical lymphocytes are more frequently observed [30,31]. The WHO-EORTC classification lists several clinical, pathological, and immunophenotypic variants of MF, with only three included: folliculotropic MF, pagetoid reticulosis, and granulomatous slack [1].

### 2.2. Immunohistochemistry of Mycosis Fungoides

The typical immunophenotype of patch-stage MF is ßF1+/CD3+/CD4+/CD8−/TIA-1−, usually with a proportion of reactive CD8+ lymphocytes; about 10% of cases may present either a true cytotoxic phenotype (ßF1+/CD3+/CD8+/CD4−/TIA-1+), or a gamma-delta phenotype (ßF1−/CD3+/CD56+/TIA-1+) [1,32,33,34]. Malignant T-cells from MF skin express the CCR4+/CLA+/CCR7−/L-selectin− immunophenotype of the skin resident memory T-cells. In contrast, clonal malignant T-cells from the blood of SS patients express the CCR4+/CCR7+/L-selectin+ phenotype of the skin-tropic central memory T-cell. The expression of the surface ligand CCR4 determines the T-cell skin homing ability, while CCR7/L-selectin controls their capacity to recirculate between the blood and lymph node. High CCR4 expression is consistently observed in advanced MF [1,32,35,36].

It is crucial to emphasize that the clinical diagnosis of MF and its variants always relies on a comprehensive clinicopathologic correlation. For detailed information on MF variants in both adults and children, it is recommended to refer to specific reviews [32,37,38].

### 2.3. Histopathology of Tumor Stage of Mycosis Fungoides and CD30 Antigen

Tumors of MF often exibit a nodular infiltrate in the dermis, and in some cases, it may extend to subcutaneous tissue. Epidermotropism, the tendency of neoplastic lymphocytes to infiltrate the epidermis, may be lost. The neoplastic lymphocytes can vary in appearance, with some showing characteristics of immunoblastic cells, including round to oval nuclei with prominent central nucleoli. Others may have lobulated or cerebriform nuclei, or they may display abundant cytoplasm, prominent nucleoli, and occasionally resemble Reed–Sternberg cells. Large cell transformation (LCT), defined as the presence of large cells in more than 25% of the infiltrate, occurs in approximately 50% of tumoral MF cases. LCT is typically associated with the expression of CD30 and GATA3, while it is negative for *IRF4* translocations (Figure 1) [3,32].

Interestingly, CD30 expression can be observed not only in tumors of MF but also in earlier stages, such as patch- or plaque-stage MF, as well as in SS [39].

CD30, previously known as Ki-1, is a cell membrane protein of the tumor necrosis factor receptor family. It is expressed by non-neoplastic activated T- and B-cells but also serves as an important tumor marker [3,32,40]. Consequently, CD30 is expressed in a wide range of reactive skin conditions (i.e., arthropod byte, molluscum contagiosum, etc.) [41], and notably in several systemic and cutaneous lymphomas [42]. In addition to tumor MF, CD30 is typically found in Hodgkin lymphoma, both systemic and pcALCL, and LyP. Aberrant CD30 positivity is also observed in other skin lymphomas, such as certain CBCL subtypes, and lymphoproliferative disorders (LD) in HIV-positive patients [3,32,42].

### 2.4. Histopathology and Immunohistochemistry of Lymphomatoid Papulosis

While clinical presentation is often characteristic and typical, the histopathology of LyP can be variable. This variability arises because the distribution of the pattern of the infiltrate may differ, leading to several histopathological variants designated with alphabet letters from A to I. In addition to common features, such as the presence of large atypical lymphocytes and the expression of CD30, each histopathological variant of LyP may resemble either a reactive skin condition (e.g., Type A: arthropod reaction) or another lymphoma (e.g., Type B: MF; Type C: ALCL; Type D: aggressive epidermotropic CD8+ cytotoxic TCL; Type E: aggressive “angiotropic” lymphomas). The immunophenotype is typically CD4+, but some variants (Type D, Type E, and Type G) exhibit a CD8+ phenotype. Therefore, as with all CTCLs, clinical–pathologic correlation is essential for an accurate diagnosis. Interestingly, various LyP variants may coexist in one patient simultaneously, and the histopathologic “type” does not carry prognostic implications [3,32,43].

### 2.5. Histopathology and Immunohistochemistry of Primary Cutaneous Anaplastic Large T-Cell Lymphoma

Conventional pcALCL typically presents as a nodular or diffuse infiltrate in the dermis, extending into the subcutis. It is characterized by cohesive sheets of large CD30+ atypical cells with rounded or irregularly shaped nuclei, prominent nucleoli, abundant cytoplasm, and, in some cases, giant cells reminiscent of Reed–Sternberg cells. Prominent necrosis and abundant eosinophils are frequently observed [3,32]. 

Similarly to Lyp, pcALCL exhibits several morphological and histopathological variants [44].

The presence of CD30+ cells should exceed 75% of the infiltrate. The most common, though not exclusive, immunophenotype is CD3+, CD4+, CD8−, CD15−, EMA−, MUM-1+, GATA3−. ALK expression is typically absent in pcALCL, but it is a characteristic feature of systemic ALCL [3,32,45]. However, ALK+ cases of pcALCL are possible and appear to be more common in children and adolescents [46].

Recurrent t (6;7) (p25.3;q32.3) translocations, involving the *DUSP22–IRF4* locus on 6p25.3, have been identified in ALK-negative ALCL, including cutaneous cases. This translocation is expressed in a specific LyP variant but is rarely present in MF with LCT, serving as a potential diagnostic criterion [32,47].

Nevertheless, the diagnosis of pcALCL should only be made after thorough staging investigations, clinical history, and clinical examination to exclude the presence of tumor-stage MF, LyP Type C, and systemic ALCL. For a detailed description of all LyP and ALCL variants, please refer to specific reviews [32,43,48].

### 2.6. Histopathology and Immunohistochemistry of Sezary Syndrome

The histopathology of Sézary syndrome (SS) closely resembles that of patch- or plaque-stage MF, but with less or even absent epidermotropism. As a result, the differential diagnosis between SS and MF, as well as other forms of erythroderma, can only be established through clinical–pathologic correlation. In particular, cytofluorimetric data are essential for detecting and quantifying blood involvement in SS. The immunophenotype typically consists of CD3+, CD4+, CD7−, and CD8−, and CD30 may be expressed by both early lesions and LCT (Figure 2) [3,32]. 

Recent studies have identified new biomarkers, including PD-1 (CD279) and KIRDL2 (CD158k), which can assist in distinguishing Sézary syndrome (SS) from reactive forms of erythroderma. Strong positivity for PD-1 in both epidermal and band-like infiltrate cells favors a diagnosis of SS, although PD-1+ cells can also be found in MF [3,16,32]. PD-1, PDL-1, and ICOS expression have been observed in all stages of MF. Specifically, ICOS or PD-L1 expression is associated with advanced-stage disease and LCT [49,50].

As a result, in both MF and SS, in addition to CD30 (targeted by brentuximab vedotin), several new immunohistochemical markers play a crucial role in assessing the feasibility of specific treatments. These markers include CD52 (targeted by alemtuzumab), CCR4 (targeted by mogamulizumab), PD-1 (targeted by nivolumab), PDL-1 (targeted by pembrolizumab), CD70 (targeted by cusatuzumab), and KIRDL2 (targeted by lacutamab) [16,17].

### 2.7. Histopathology and Immunohistochemistry of Blastic Plasmacytoid Dendritic Cell Neoplasm

BPDCN is characterized by histopathological findings that include a diffuse dermal and subcutaneous infiltrate comprising monomorphous medium-sized atypical cells with a blastic morphology. Epidermotropism is absent, and there is a distinct grenz zone that separates the epidermis from the dermal infiltrate. Subcutaneous involvement is common, and hemorrhages are typically observed [5,6,9,10,32].

BPDCN cells express CD4, CD56, CD123, and in 30–60% of neoplastic cells, TdT is positive, along with Ki67 (in 20 to 90% of the cells) and CD68. Blastic cells must be negative for NK-cell markers, cytotoxic markers (such as granzyme B), myeloperoxidase, and lysozyme, as well as the myeloid cell nuclear differentiation marker. Other significant positive markers of BPDCN include T-cell leukemia/lymphoma (TCL)-1, blood dendritic cell antigen (BDCA-2, CD303), CD304, and the transcription factor TCF4. Rare cases of BPDCN that are negative for CD4, CD56, and even CD123 have been reported [5,6,9,10,32,51].

Notably, CD123 is the most important marker, as it is not only diagnostic but also serves as the target for tagraxofusp [11,52].

## 3. Brentuximab Vedotin

CD30, also called TNFRSF8 or Ki-1, is a transmembrane glycoprotein receptor with intracellular, trans-membrane, and extracellular domains. It belongs to the tumor necrosis factor receptor (TNFR) family. CD30 is expressed in a small number of activated T and B-lymphocytes and is also found in several lymphoid neoplasms, particularly classical Hodgkin lymphoma (HL) and ALCL. It can be also found in cutaneous T-cell lymphoma (CTCL). The actions of CD30 are mediated by various signaling pathways that promote the survival of the cells on which CD30 is upregulated [18]. 

CD30 activation leads to the trimerization of tumor necrosis factor receptor-associated proteins (TRAF), particularly TRAF2, TRAF1, and TRAF5, resulting in the activation of the NF-kB pathway. Additionally, TRAF2 activates the mitogen-activated protein kinase/extracellular signal-regulated kinases (MAPK/ERK) pathways. These two activated pathways enhance anti-apoptotic and pro-survival effects in the neoplastic cell. Furthermore, a self-perpetuating positive feedback loop through the MAPK/ERK pathway and the expression of JunB, a member of the activator protein (AP-1) nuclear transcription factors, amplifies these effects by upregulating the expression of CD30 on the cell. The upregulation of CD30 in numerous neoplasms makes it a key therapeutic target [18].

Brentuximab vedotin (BV) is an anti-CD30 humanized IgG monoclonal antibody combined with the antimitotic agent monomethylauristatin E (MMAE) via a cathepsin cleavable linker (valine-citrulline). The randomized phase III ALCANZA trial approved BV for the treatment of CTCL [15].

The binding of antibody-drug conjugate (ADC) to CD30-positive cells lead to receptor endocytosis, resulting in the release of MMAE when exposed to intracellular lysozymes. This cascade leads to the inhibition of tubulin formation and cell apoptosis [18].

The BV standard therapeutic schedule is 1.8 mg/kg iv over 30 min every 3 weeks for 16 cycles or until unacceptable toxicity or disease progression. The randomized trial ALCANZA compared BV with physician’s choice of either bexarotene or methotrexate in patients with CD30-positive MF or pcALCL having received at least one prior systemic therapy or radiotherapy. Patients with SS were not included. CD30 positivity was defined as having at least one biopsy with ≥10% of target lymphoid cells exhibiting a membrane and/or intracellular staining pattern for CD30 [15]. 

The BV group received 1.8 mg/kg IV every 3 weeks for up to 16 cycles, while the physician’s choice group took methotrexate (5–50 mg orally once weekly) or bexarotene (300 mg/m^2^ orally once daily). Unless there was disease progression or intolerable toxicity, the maximum duration of treatment was 48 weeks. The results showed that BV was superior to physician’s choice for all stages of MF, with significant improvements in the overall response rate sustained for at least 4 months (ORR4: 56% vs. 13%), overall response rate (ORR: 67% vs. 20%), complete response rate (CR: 16% vs. 2%), progression-free survival (PFS: 16.7 months vs. 3.5 months), and 3-years overall survival rate (OS: 64.4% vs. 61.9%). With a median follow-up of 37.3 months, time to next treatment (TTNT) was significantly longer with BV (13.4 months) compared to that with physician’s choice (5.6 months) [15].

The ALCANZA sub-analysis revealed that the greatest ORR4 and PFS were observed in patients with CD30-positive MF and at least one biopsy showing ≥10% CD30 expression, regardless of LCT status [53].

Peripheral neuropathy (PN) was a common side effect in the BV arm, affecting 66% of patients overall, with 9% experiencing grade 3 PN, leading to treatment discontinuation in 14% of cases [15,54]. Sensory peripheral neuropathy is the most frequent adverse event, occurring in 45–67% of patients, typically graded as 1–2, but cases of grade 3 and 4 have been reported. It often presents as distal numbness, vibratory sensory loss, gait alteration, and abnormality of tactile discrimination [55,56]. This manifestation seems to be due to off-target effects of BV, in particular the anti-tubulin effects of MMAE. PN generally resolves with dose reduction, although in some cases, treatment discontinuation may be necessary. Corbin et al. observed a median timing of 43 weeks for improvement or resolution of PN from the last BV dose [56].

Fatigue is another common side effect reported by 29–47% of patients, followed by complaint of gastrointestinal alterations. Other less frequent advent events include hematologic abnormalities (cytopenia, neutropenia, and anemia), acute pancreatitis, and demyelinating polyneuropathy [56].

Due to the potential for embryo-fetal toxicity, BV should be avoided by women of childbearing age who wish to become pregnant [55].

Moreover, BV has demonstrated rapid effectiveness in LCT MF with CD30 expression, serving as a bridge to allogeneic stem cell transplantation (ASCT), and it can be used during the resumption after ASCT in case of disease recurrence [57]. 

## 4. Mogamulizumab 

C–C chemokine receptor 4 (CCR4) is a transmembrane receptor for chemokines CCL17 and CCL22 belonging to transmembrane metabotropic receptors family. It plays a critical role in cell trafficking of lymphocytes [58]. Its expression on regulatory dendritic cells, neurons, microglia, and astroglia explains the involvement of the receptor in the pathogenesis of autoimmune encephalitis, multiple sclerosis, asthma, and atopic dermatitis. Furthermore, CCR4 is found on various immune cell types, including T-cells, Th2, Th17, natural killer cells, macrophages, monocytes, and neoplastic T-cells, including those in adult T-cell leukemia/lymphoma, peripheral T-cell lymphoma, and CTCL [58].

Mogamulizumab is a humanized IgG1 monoclonal antibody directed against CCR4. Binding of CCR4 and the afucosylated FC region of the antibody leads to an increase of antibody-dependent cellular cytotoxicity (ADCC) activity. Mogamulizumab eliminates both tumor cells and regulatory T-cells (Tregs) that express CCR4 on the cell surface, enhancing its antineoplastic efficacy [19].

In the US and EU, the open-label randomized phase III MAVORIC trial resulted in the approval of mogamulizumab for adult patients with relapsed or refractory MF and SS after failure of at least one prior systemic therapy. Selected patients with relapsed or refractory disease after one systemic therapy were randomized 1:1 to receive mogamulizumab (1.0 mg/kg iv once weekly for the first 28-day cycle, then on days 1 and 15 of subsequent cycles) or oral vorinostat (400 mg daily) [14].

Mogamulizumab showed the most striking results in patients with blood and/or skin involvement, with a less robust response in nodal and visceral disease [59].

Mogamulizumab significantly improved progression-free survival (PFS) compared to that with vorinostat (median 7.7 vs. 3.1 months). The overall response rate (ORR) was also remarkably improved with mogamulizumab versus vorinostat (28% vs. 5%) [14]. The activity of mogamulizumab vs. vorinostat was faster in the blood (1.1 months vs. 1.9 months) than in the skin and lymph node compartments, where a longer time to response was observed (3.0 months vs. 2.7 months for skin, 3.3 months vs. 2.9 months for lymph nodes) [14].

A post hoc analysis by Cowan et al. showed that compared to vorinostat, mogamulizumab yielded superior results in SS and MF patients with B1 and B2 blood involvement. Specifically, in mogamulizumab B1 and B2 patients, longer PFS and time to next treatment (TTNT) were observed, in addition to better skin response (evaluated with modified Severity-Weighted Assessment Tool, mSWAT). Furthermore, in all blood classification groups, ORR was higher with mogamulizumab than with vorinostat, and this difference became more pronounced with increasing B class (B2 > B1 > B0) [60].

Regarding the duration of response by compartment, patients receiving mogamulizumab had a median duration of response in skin of 20.6 months (vs. 10.7 months), in blood of 25.5 months, and in lymph nodes of 15.5 months (median duration of response in blood and lymph nodes was not estimatable for vorinostat) [14]. There was evidence of a longer duration response in patients with SS despite drug discontinuation due to autoimmune manifestations [61].

The most frequent AEs with mogamulizumab were infusion-related reactions (33%), mostly grade 1 and limited to the first one or two infusions. Other common adverse events were skin rash (28%), diarrhea (23%), and fatigue (23%) [14].

Skin rash associated with mogamulizumab is now termed mogamulizumab-associated rash (MAR), and further studies have reported an even higher incidence, with rates of up to 68% in treated patients. Patients with SS tend to develop MAR more frequently than those with MF [62,63,64]. The depletion of regulatory T-cells (Tregs) may be responsible for MAR. Clinically, MAR may manifest as lichenoid eruptions, psoriasiform eruptions, photodistributed rash, MF-like patches and plaques, or SS-like erythroderma. Biopsies of MAR may show four main epidermotropic patterns: spongiotic, lichenoid, psoriasiform, and interface dermatitis. Eosinophils are frequently present, along with bi- and multinucleated cells, and epithelioid granulomas are observed in about 30% of cases [62,63]. 

The management of MAR depends on the extent of skin involvement and may be treated with topical or systemic steroids. While the current product label for mogamulizumab advises temporary interruption in the case of moderate or severe adverse events, some authors suggest that better outcomes may be achieved by continuing mogamulizumab at the same dose despite MAR, rather than temporarily discontinuing treatment [62,63]. Optimal management requires a multidisciplinary approach involving hematologists, dermatologists, and pathologists [63]. It has been reported that MAR occurred in patients with SS and was associated with a longer time to progression [65,66]. Jfri et al. described a positive association of MAR with overall responses and T4-stage (erythroderma) at the time of treatment, while it was not related to disease stage (I–IV) or subtype (MF vs. SS) [67].

The most observed grade 3–4 AEs in the mogamulizumab group were pyrexia in eight individuals (4%) and cellulitis in five subjects (3%). Three mogamulizumab patients died during treatment and two of these deaths (due to sepsis and polymyositis) were considered treatment-related [14]. The action of mogamulizumab causes lymphopenia and therefore immunosuppression, which justifies hepatitis B screening before treatment and the use of antipneumocystis and antiviral prophylaxies [21].

Based on the MAVORIC trial, mogamulizumab is indicated particularly in older fit patients (ECOG Performance Status ≤ 1) with appropriate renal, hepatic, and hematologic function [59]. However, particular attention must be paid to young subjects who are planning to undergo allogeneic stem cell transplantation because an increased risk of graft-versus-host disease (GVHD) has been observed if transplantation is performed shortly after mogamulizumab bridging therapy [68]. Mogamulizumab can reduce normal Tregs, increasing the risk of GVHD in patients who undergo ASCT if mogamulizumab treatment was administered <50 days before transplantation. In such cases, it is important to carefully assess the timing and pre-transplant prophylaxis interventions [68].

Blackmon et al. emphasized the need to avoid the use of mogamulizumab in patients with pre-existing autoimmune diseases due to the possible development of serious AE such as hypothyroidism, pneumonitis, hepatitis, immune mediated myositis, myocarditis, Guillain-Barré syndrome, polymyositis and polymyalgia rheumatica [59,69]. These events appear to be related to the potential reduction of CCR4-expressing Tregs cells, resulting in the development of immune reactions against off-targets in the context of autoimmunity [59]. 

According to the Jouandet et al. retrospective real-life study on the efficacy and tolerability of mogamulizumab, PFS was estimated to be 22 months. The study also observed good tolerability with few grade 3–4 adverse events, and the median time between the administration of mogamulizumab and the occurrence of the first adverse event was 21 days [70].

A large French study conducted by Beylot-Barry et al. included 122 patients (69 with SS and 53 with MF) and confirmed the efficacy and tolerability of mogamulizumab in multifailure patients with advanced disease (stage IIB–IVB). Notably, a high response in the blood compartment (81.8% of SS patients) was observed [71].

It should be emphasized that the excellent tolerability profile of mogamulizumab makes the drug very attractive for designing new combination studies with other therapies [72]. The combination of mogamulizumab with TSEB has shown very promising clinical efficacy in some patients with CTCL [73]. An open-label, multi-center, phase II trial (MOGAT) study in patients with early-stage CTCL is ongoing [74].

The multicenter phase III trial conducted by Porcu et al. analyzed the quality-of-life effects of mogamulizumab compared to those of vorinostat in patients with stage IB–IV MF/SS who had failed at least one systemic therapy. The results demonstrated that patients treated with mogamulizumab, especially those with worse symptoms and functional impairment, experienced greater benefits in terms of symptom reduction, improved functionality, and an overall increase in the quality of life [75].

Horwitz et al. conducted a post hoc analysis based on the MAVORIC trial to investigate the impact of therapies performed prior to mogamulizumab considering type, number, and time lapse between last therapy and mogamulizumab. The results indicate that prior therapies do not change ORR, PFS, and duration of response (DOR). Furthermore, the time elapsed from prior therapy and its immunomodulatory activity do not impact ORR and PFS evaluated after mogamulizumab treatment [76]. 

## 5. Anti-PD1 and Anti-PDL1 (Pembrolizumab + Nivolumab + Atezolizumab) 

Programmed cell death protein 1 (PD-1) is transmembrane receptor protein found in B and T-lymphocytes, belonging to the immunoglobulin superfamily. As an immune checkpoint, it plays a crucial role in regulating the immune response by increasing tolerance towards self-antigens, thereby reducing the inflammatory activity of T-lymphocytes. When activated by its ligand, PD-L1 and PD-L2 inhibits T-cell activation and proliferation. PD-L1 and PD-L2 are frequently observed in the tumor microenvironment [22]. 

It has been noted that tumor cells in advanced CTCL often express PD-1. PD-1 or PD-L1 inhibitors, by blocking this cascade of events, allow for the activation of the immune response against neoplastic cells, primarily through CD8+ cytotoxic lymphocytes [21]. 

Pembrolizumab is a humanized monoclonal antibody targeting PD-1. In a multicenter phase II trial (CITN-10), pembrolizumab was studied in 24 patients with refractory or relapsed MF and SS (clinical stage IB to IV). The treatment regimen consisted of 2 mg/kg iv every 3 weeks for 24 months [20].

The overall response rate (ORR) was 38%, with two complete responses (CR) in SS patients and seven partial responses (five in MF and two in SS). PFS and OS rates at one year were 65% and 95%, respectively. Patients with MF fared slightly better than those with SS in terms of response rate (56% vs. 27% respectively), and long-lasting responses could be achieved [20].

Regarding immune-related AEs, grade 1–3 were the most observed, including cutaneous events (also rash and skin flare), arthritis/arthralgia, pneumonitis, gastrointestinal inflammation, and elevated AST and ALT. No grade 4–5 AEs occurred. Treatment-related AEs such as anemia, hypo-hypertension, pulmonary or periorbital edema, fever, and hypocalcemia were less frequent [20].

In patients with SS treated with PD-1 inhibitors, a transient skin flare with manageable worsening of erythema and pruritus can occur, but nevertheless, treatment should be continued with caution [20].

A predictor of this flare reaction could be increased expression of PD-1 on the surface of circulating Sézary cells (the use of flow cytometry in patients with SS could be a practical assay for predicting this phenomenon) [20].

Nivolumab is a human monoclonal antibody directed against PD-1. Its activity in MF and SS has been observed by Lesokhin et al. in a phase Ib study that included patients with relapsed or refractory T-cell lymphoma, B-cell lymphoma, or multiple myeloma. Thirteen patients with mycosis fungoides (MF), five with peripheral T-cell lymphoma (PTCL), two with Sezary syndrome CTCL, and three with other non-CTCL were included in the T-cell lymphoma group. Nivolumab 1 or 3 mg/kg was provided as a 1 h infusion at weeks 1 and 4, and then every 2 weeks for up to 2 years. In the T-cell lymphomas group (n = 23 patients), the response rate was 17%. Two MF patients (15%) had a PR, while nine (69%) had stable disease [21,22].

Drug related AEs of any grade occurred in 74% of patients of the T-cell lymphomas group, and 22% were grade ≥ 3. Skin rash and pruritus were the most frequent drug related AEs, and pneumonitis (4% grade ≥ 3), enteritis and diarrhea, hypersensitivity, blood increases of AST, ALT, and creatinine occurred more rarely. Immune-mediated AEs occurred in 34% of all patients and were mostly grade 1 or 2 [22].

An ongoing phase II trial (NCT03357224) is evaluating the use of atezolizumab (anti-PD-L1) in stage IIb-IV MF and SS relapsed/refractory after a previous systemic treatment [23]. 

## 6. Alemtuzumab

CD52 is a membrane glycoprotein found in the majority of normal peripheral T and B-cells, as well as in most malignant B- and T-lymphocytes.

Alemtuzumab is an anti-CD52 humanized monoclonal antibody that specifically targets CD52-positive cells. It works by causing the depletion of T-cells, B-cells, as well as NK (natural killer) cells and monocytes, leading to cell death through antibody-dependent cellular cytotoxicity and complement-dependent cytotolysis. Several studies have demonstrated the efficacy of alemtuzumab, particularly in patients with refractory erythrodermic blood-involved MF/SS, where it can effectively reduce malignant cell populations [24].

Diffuse erythema is mainly induced by central memory T-cells in transit with only recirculating T-cells that enter the bloodstream being targeted by alemtuzumab. Neoplastic T-cells in blood disease are central memory cells that migrate between the skin, blood, and lymph nodes, while malignant T-cells in cutaneous-limited MF are skin-resident memory T-cells. Alemtuzumab eliminates malignant lymphocytes via neutrophils and/or NK cells which are typically found in the blood and not on the skin [77].

The standard therapeutic schedule consists of increasing doses from 3–10 mg to 30 mg administered IV every other day, followed by 30 mg three times a week for at least 12 weeks. The most frequent adverse effects for the IV schedule of alemtuzumab are infections and hematologic toxicity [24].

Lundin et al. evaluated the use of alemtuzumab in low-grade non-Hodgkin’s lymphomas that had been previously treated with chemotherapy. In this study, four of eight patients with mycosis fungoides (50%) responded to alemtuzumab with two CRs and two PRs [78]. Based on this promising experience, a phase II study was conducted to investigate alemtuzumab as a single agent. This study included eight patients with relapsed or refractory advanced mycosis fungoides (MF) or Sézary syndrome (SS), with seven having MF/SS and one with large-cell transformation. ORR was 38% (with three PRs, two stable disease, and three progressive disease), and greater efficacy was shown in patients with erythroderma (69% ORR) than in those with plaques or skin tumors (40% ORR). All three patients with circulating neoplastic cells pre-alemtuzumab had blood clearance. All patients developed progressive disease (PD) within 4 months of starting therapy. Significant hematological and immunosuppressive toxicity was highlighted [79]. Considering the marginal efficacy and substantial side effects, there was interest in exploring combination therapeutic regimens. Weder et al. reported a case of familiar cutaneous MF involving a father and son who achieved a remarkable skin response and resolution of enlarged lymph nodes with a combination of gemcitabine and alemtuzumab. Importantly, this combination therapy was well tolerated and did not lead to increased toxicity [80]. 

A multicenter retrospective analysis was conducted involving 39 patients with SS or advanced MF who were treated with alemtuzumab, with the aim of evaluating the treatment’s long-term efficacy and safety. ORR was 51% (13 patients with PR and 7 patients with CR): 70% in SS patients and 25% in MF patients. It is important to highlight that during alemtuzumab treatment, five patients developed cutaneous large T-cell transformation. Several grade 3 or greater infectious AEs were observed. What can be deduced from the results is that alemtuzumab would appear to induce long-term remission in SS, but its efficacy is drastically reduced in MF and transformed CTCL [25,77].

Additionally, the use of low-dose intermittent alemtuzumab, administered subcutaneously (10–15 mg on alternating days) was described by Bernengo et al. in a study involving 14 patients with relapsed/refractory SS who had high counts of circulating Sézary cells. This study demonstrated a favorable toxicity profile, a high response rate, and durable remission, with the lower dose (10 mg) appearing to be more effective. Furthermore, the subcutaneous administration schedule was associated with reduced toxicity when compared to the intravenous route, and a lower incidence of infusion-related adverse events [81].

## 7. Lacutamab

The KIR3DL2 (CD158K) receptor belongs to the KIR (Killer Immunoglobulin-like Receptors) superfamily, and its ligands are class I MHC molecules. Normally expressed in NK lymphocytes and part of cytotoxic T-cells, it is also presented by neoplastic CD4+ T-cells of SS patients, transformed MF, CD30+ cutaneous lymphomas, and other uncommon subtypes. Most KIR proteins transmit inhibitory signals that inhibit the cytotoxic action of the NK cell on which they are expressed [26].

Lacutamab is a first-in-class humanized monoclonal antibody targeting KIR3DL2, first studied in a phase I clinical trial including 42 patients with relapsed/refractory primary CTCL comprising 35 cases of Sézary syndrome (SS), 8 cases of mycosis fungoides (MF), and 1 case of primary CTCL not otherwise specified. Lacutamab was administered intravenously weekly for 5 weeks, followed by administration every 2 weeks for a total of 10 administrations, and then transitioned to every 4 weeks, continuing until disease progression or the occurrence of unacceptable toxicity [26].

After a 14-month follow-up, an ORR of 36% was seen, with 43% of those having SS. The most common AEs were grade 1 or 2 peripheral edema (27%) and fatigue (20%). The most frequent grade 3 adverse event was lymphopenia [26].

An ongoing multi-cohort phase II clinical trial (Tellomak, NCT03902184) is currently testing lacutamab in MF/SS patients without large cell transformation [21]. 

## 8. Cusatuzumab 

Cusatuzumab is a CD70-directed monoclonal antibody. CD70 is predominantly expressed on highly activated T-cells and B-cells, but it is also found on NK cells and mature dendritic cells. It plays a critical role in the control of immune system activation, particulary in enhancing T-cell and B-cell activation, proliferation, and survival, which leads to a more effective immunological response.

Cusatuzumab blocks CD70-CD27 signaling, stopping the proliferation of leukemic stem cells. Additionally, it activates Fc-mediated cytoxicity and promotes antibody-dependent cellular toxicity (ADCC) against neoplastic cells [27]. 

A phase I/II cohort expansion study (NCT01813539) enrolled 27 patients with R/R cutaneous T-cell lymphomas who received cusatuzumab intravenously at 1 mg/kg (n = 11) or 5 mg/kg (n = 16) once every 3 weeks. The most common AEs included infusion-related reactions, pyrexia, and asthenia [27]. 

The mean treatment duration was 5.2 months. The overall response rate was 23%, comprising one complete response (CR) and five partial responses (PRs), with nine patients achieving stable disease (SD). Notably, patients with Sézary syndrome (SS) demonstrated a 50% ORR at both doses, with a 60% PR rate at a dosage of 5 mg/kg and a 33% PR rate at a dosage of 1 mg/kg. Considering the dose-dependent effect, a dosage of 5 mg/kg can be considered for future development [27]. 

## 9. Tagraxofusp 

Tagraxofusp (SL-401) is a recombinant fusion protein composed of human interleukin-3 fused to truncated diphtheria toxin, targeting the interleukin-3 receptor (CD123). The interleukin-3 portion binds to CD123, facilitating the drug’s entry into cells, where the diphtheria toxin is released, leading to cell death. CD123 is typically found on the surface of basophils, eosinophils, monocytes, myeloid dendritic cells, and plasmacytoid dendritic cells, regulating the cells’ ability to reproduce and become more specialized [28].

Tagraxofusp, approved by the FDA and EU for use in patients diagnosed with BPDCN, is given by intravenous infusion. In BPDCN, neoplastic cells express the interleukin-3 receptor at a high level on their surface. Pemmaraju et al. conducted a study to prospectively evaluate the efficacy of tagraxofusp in patients with untreated or previously treated and relapsed BPDCN. The dose selected after the dose-finding stages of the study was 12 μg per kilogram of body weight, administered on days 1 to 5 of each 21-day cycle [29].

This prospective study, the largest to date, evaluating CD123-targeted therapy in BPDCN, was recently updated with long-term results including 89 patients. Among the participants, 84 out of 89 patients received TAG at a dose of 12 mcg/kg once daily. The results have shown an overall response rate (ORR) of 75% in first-line (1L) and 58% in relapsed/refractory (R/R) patients. In untreated patients (65 out of 84), the complete response (CR) + clinical CR (CRc; CR with residual skin abnormality not indicative of active disease) rate was 57%, with 51% of these patients bridging to hematopoietic stem cell transplant (HSCT). The median duration of CR + CRc was 25 months as observed during 3 years of follow-up. In R/R patients (19 out of 84), the CR + CRc rate was 16%, with 5% of patients bridging to HSCT [29].

The most common side effects of tagraxofusp include fatigue, fever, nausea and diarrhea, elevated levels of alanine aminotransferase and aspartate aminotransferase, hypoalbuminemia, and thrombocytopenia. Severe AEs could include hepatic dysfunction, thrombocytopenia, and capillary leak syndrome (CLS). The latter consists of endothelitis, with fluid leakage from the intravascular compartment into surrounding tissue resulting in lower blood pressure and weight gain. CLS was reported in 19% of the patients and was associated with two deaths, one for each dose group [29]. In the long-term follow-up cohort, CLS was reported in 21% of cases, with a well-characterized and manageable safety profile, with early intervention guidelines [29].

## Figures and Tables

**Figure 1 cells-12-02656-f001:**
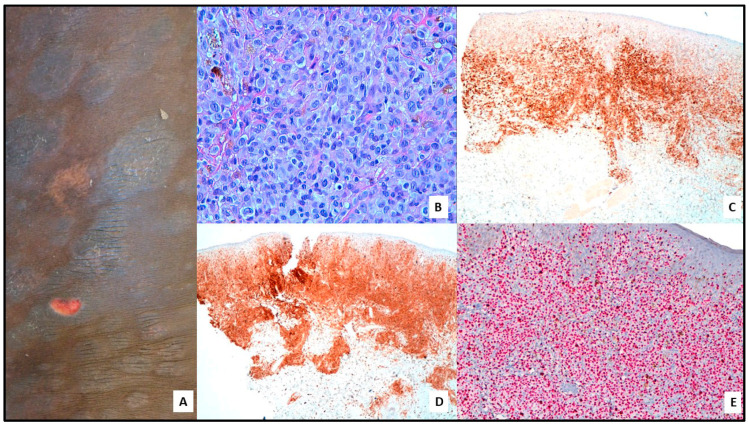
(**A**) MF with LCT: multiple plaques and nodules, partly ulcerated on the trunk. (**B**) Large neoplastic lymphocytes, partly immunoblastic in appearance with round to oval nuclei and prominent central nucleoli. (**C**) Strong CD30 positivity (40×). (**D**) Strong CD4 positivity (10×). (**E**) High Ki67 expression (100×).

**Figure 2 cells-12-02656-f002:**
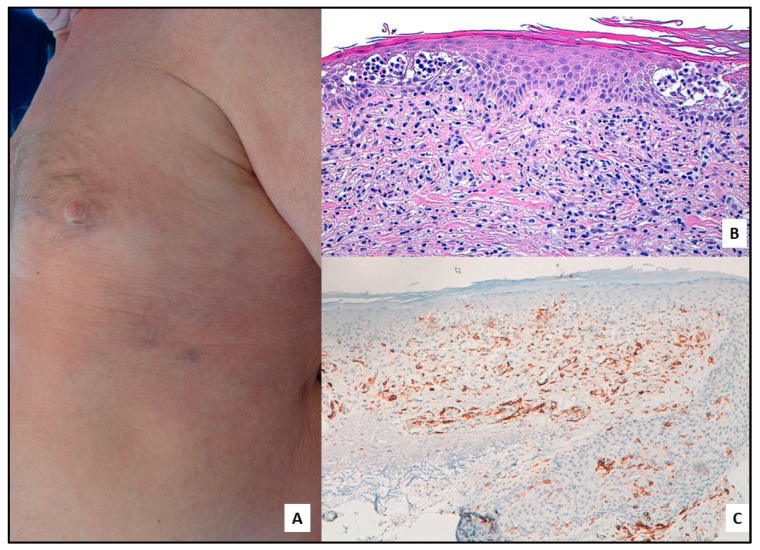
(**A**) SS: diffuse erythema on the trunk. (**B**) Atypical epidermotropic lymphocytes mainly collected in the typical Pautrier’s (Darier’s) microabscess; band-like infiltrate in the superficial dermis (H&E; 40×). (**C**) Strong CD30 positivity (40×).

**Table 1 cells-12-02656-t001:** Available therapies for MF, SS, ALCL, LyP, and BPDCN. CTCL = cutaneous T-cell lymphomas; MF = mycosis fungoides; SS = Sezary syndrome; Q3w = every 3 weeks; mths = months; R/R = relapsing/remitting; qw = once a week; Q2w = every 2 weeks; q.o.d. = every other day; t.i.w. = 3 times a week; Q1w = every week; Q4w = every 4 weeks; q.d. = once a day; BPDCN = blastic plasmocytoid dendritic cell neoplasm.

Drug	Target	Disease	Treatment Schedule	ORR	PFS	References
Brentuximab vedotin	CD-30	CD30+ CTCL after at least 1 prior systemic therapy	1.8 mg/kg iv Q3w for 16 cycles	67%	16.7 mths	[15,18]
Mogamulizumab	CCR-4	R/R MF and SS	1.0 mg/kg iv qw for the first 28-days cycle, then on days 1 and 15 of subsequent cycles	28%	7.7 mths	[14,19]
Pembrolizumab	PD-1	R/R MF and SS	2 mg/kg iv Q3w for 24 months	38%	65% at 1 year	[20]
Nivolumab	PD-1	R/R CTCL	1 or 3 mg/kg at weeks 1 and 4, then Q2w for up to 2 years	-	10 mths	[21,22]
Atezolizumab	PD-L1	R/R stage IIb-IV MF and SS	1200 mg IV Q3w for 1 year	ongoing study	ongoing study	[23]
Alemtuzumab	CD-52	stage IIb-IV MF and SS	increasing doses from 3–10 mg to 30 mg i.v. q.o.d., followed by 30 mg t.i.w. for at least 12 weeks	51%	56 mths	[24,25]
Lacutamab	KIR3DL2	R/R CTCL	increasing doses from 0.0001 mg/kg to 10 mg/kg Q1w for the first month, Q2w for 10 doses, then Q4w	36.40%	8.2 mths	[26]
Cusantuzumab	CD70-CD27	R/R CTCL	1 mg/kg or 5 mg/kg Q3w	23%	-	[27]
Tagraxofusp	CD123	BPDCN	12 mcg/kg q.d. on days 1 to 5 of each 21-day cycle	75% as first-line 58% in R/R patients	-	[28,29]

## Data Availability

Data sharing not applicable. No new data were created or analyzed in this study. Data sharing is not applicable to this article.

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
