# Peer review of "Histopathological Markers for Target Therapies in Primary Cutaneous Lymphomas"

_cells, 2023, doi:10.3390/cells12222656_

Round 1

Reviewer 1 Report

Comments and Suggestions for Authors

This article provides an overview of a study that aims to provide current insights and information on the latest advances in treatment approaches for cutaneous primary cell lymphomas (PCTCLs). Special emphasis is placed on the recent emergence and ongoing evaluation of targeted (biological) therapies. Furthermore, it highlights the increasing importance of immunohistochemical staining in skin biopsies and cytofluorimetry, which plays a fundamental role in informed decision-making for the management of patients with PCTCL. The primary goal of this research is to increase understanding of the evolving treatment landscape associated with this particular type of lymphoma and to contribute to the overall improvement of care for people with PCTCL.

Despite the encouraging findings, the reviewer recommends shortening the manuscript title and also emphasizes the need for a comprehensive conclusion or summary. This manuscript stands out for its novelty and significant contribution to the field. The current level of scientific review evident in the manuscript is considered sufficient for publication in this particular journal with minor revisions. After thorough evaluation, my final recommendation for this manuscript is to minor review.

Author Response

We thank the reviewer for this positive comment.

The title has been shortened.

To summarize the content of the article, a summary table (Table 1) containing all the therapies described in the manuscript has been added.

All corrections made have been highlighted in yellow in the text.

Last, we would like to underline that we have removed Figure 1 from the main text in order to create a graphic abstract (GA), as suggested.

Reviewer 2 Report

Comments and Suggestions for Authors

This interesting review provides an overview of primary cutaneous lymphomas (PCTL) and current therapeutic strategies. Emphasis is placed on mycosis fungoides (MF), representing about 60% of CTCL. While the work presented here is interesting, it presents multiple limitations that should be adequately addressed before the publication in Cells journal. Specifically, I have the following concerns:

Major points:

1.     The authors should extensively review the English language, typos, general mistakes throughout all the manuscript. Different sections are unclear, ultimately impairing the general flow and readiness of the draft. Abstract also needs to be improved.

2.     Page 3 paragraph 2 the authors say: “MF is the prototypic type of an epidermotropic lymphoma. Neoplastic lymphocytes are present in the epidermis in 96% of biopsies of patch stage MF (more frequently in single unit, in about only 20% of cases collected in the typical Pautrier’s (Darier’s) microabscess); atypical lymphocytes should be observed. Usually a band-like or patchy lichenoid infiltrate is present in the superficial dermis. Plaque stage MF present similar features with more pronounced epidermal hyperplasia and a denser infiltrate”. This sentence is unclear and confusing.

3.     Please, include subheadings in the paragraph 2 to make the reading clearer and more fluid.

Minor points:

1.     Please, add references where necessary throughout the entire manuscript, particularly in paragraphs 3, 4, 5, and 6.

2.     Page 11 paragraph 6: please substitute “complement-depentent cytotoxicity” with “complement-dependent cytolysis”.

3.     Figure letters are hardly visible, please adjust.

4.     In the abstract substitute “expression of these antibodies” with “expression of these epitopes”.

Comments on the Quality of English Language

T  The authors should extensively review the English language, typos, general mistakes throughout all the manuscript. Different sections are unclear, ultimately impairing the general flow and readiness of the draft. Abstract also needs to be improved.

Author Response

We thank the reviewer for the comment.

Major points:

  1. The authors should extensively review the English language, typos, general mistakes throughout all the manuscript. Different sections are unclear, ultimately impairing the general flow and readiness of the draft. Abstract also needs to be improved.

 R: Thank you for the comment. A careful linguistic and grammatical revision has been performed to make the text more comprehensible and fluent.

  1. Page 3 paragraph 2 the authors say: “MF is the prototypic type of an epidermotropic lymphoma. Neoplastic lymphocytes are present in the epidermis in 96% of biopsies of patch stage MF (more frequently in single unit, in about only 20% of cases collected in the typical Pautrier’s (Darier’s) microabscess); atypical lymphocytes should be observed. Usually a band-like or patchy lichenoid infiltrate is present in the superficial dermis. Plaque stage MF present similar features with more pronounced epidermal hyperplasia and a denser infiltrate”. This sentence is unclear and confusing.

 R: We thank the reviewer for this comment. The paragraph has been reformulated in this way: "MF is the prototypic type of an epidermotropic lymphoma. Neoplastic lymphocytes are present in the epidermis in 96% of biopsies in the patch stage of MF. Epidermotropic lymphocytes are found more frequently as single units through the epidermis or along the basal layer. In about only 20% of cases, they are collected in the typical Pautrier's (Darier's) microabscess. Atypical lymphocytes are diagnostic but are rarely seen in early MF (about 10% of biopsies). Interestingly, a band-like (30%) or patchy lichenoid infiltrate (66%) is present in the superficial dermis. Plaque stage MF presents similar features to early MF, but the epidermal hyperplasia and the dermal infiltrate are more pronounced. Moreover, Pautrier's (Darier's) microabscess and atypical lymphocytes are more frequently observed [18, 20]. Several clinical, pathological, and immunophenotypic variants of MF have been described, with only three listed in the WHO-EORTC classification: folliculotropic MF, pagetoid reticulosis, and granulomatous slack [1]."

  1. Please, include subheadings in the paragraph 2 to make the reading clearer and more fluid.

 R: We thank the reviewer for this comment. As suggested, seven subheadings have been added to enhance the paragraph's organization.

Minor points:

1. Please, add references where necessary throughout the entire manuscript, particularly in paragraphs 3, 4, 5, and 6.

 R: We thank the reviewer for this comment. Numerous references have been added, especially in the indicated paragraphs

2. Page 11 paragraph 6: please substitute “complement-depentent cytotoxicity” with “complement-dependent cytolysis”.

R: We thank the reviewer for this suggestion; the substitution has been made.

3. Figure letters are hardly visible, please adjust.

R: We thank the reviewer for this comment. The figures have been modified so that the letters are more visible

4. In the abstract substitute “expression of these antibodies” with “expression of these epitopes”.

R: We thank the reviewer for this suggestion; the substitution has been made.

All corrections made have been highlighted in yellow in the text.

Last, we would like to underline that we have removed Figure 1 from the main text in order to create a graphic abstract (GA), as suggested.

Round 2

Reviewer 2 Report

Comments and Suggestions for Authors

The manuscript is now suitable for the publication in the Cells journal